# The Autism Spectrum: Behavioral, Psychiatric and Genetic Associations

**DOI:** 10.3390/genes14030677

**Published:** 2023-03-09

**Authors:** Ann Genovese, Merlin G. Butler

**Affiliations:** Department of Psychiatry and Behavioral Sciences, University of Kansas Medical Center, 3901 Rainbow Blvd., MS 4015, Kansas City, KS 66160, USA; agenovese@kumc.edu

**Keywords:** autism spectrum disorder, behavior, genetic defects and associations, medical conditions, pharmacogenetics, psychiatry

## Abstract

Autism spectrum disorder (ASD) consists of a group of heterogeneous genetic neurobehavioral disorders associated with developmental impairments in social communication skills and stereotypic, rigid or repetitive behaviors. We review common behavioral, psychiatric and genetic associations related to ASD. Autism affects about 2% of children with 4:1 male-to-female ratio and a heritability estimate between 70 and 90%. The etiology of ASD involves a complex interplay between inheritance and environmental factors influenced by epigenetics. Over 800 genes and dozens of genetic syndromes are associated with ASD. Novel gene–protein interactions with pathway and molecular function analyses have identified at least three functional pathways including chromatin modeling, Wnt, Notch and other signaling pathways and metabolic disturbances involving neuronal growth and dendritic spine profiles. An estimated 50% of individuals with ASD are diagnosed with chromosome deletions or duplications (e.g., 15q11.2, BP1-BP2, 16p11.2 and 15q13.3), identified syndromes (e.g., Williams, Phelan-McDermid and Shprintzen velocardiofacial) or single gene disorders. Behavioral and psychiatric conditions in autism impacted by genetics influence clinical evaluations, counseling, diagnoses, therapeutic interventions and treatment approaches. Pharmacogenetics testing is now possible to help guide the selection of psychotropic medications to treat challenging behaviors or co-occurring psychiatric conditions commonly seen in ASD. In this review of the autism spectrum disorder, behavioral, psychiatric and genetic observations and associations relevant to the evaluation and treatment of individuals with ASD are discussed.

## 1. Introduction 

Autism spectrum disorder (ASD) is a complex neurodevelopmental condition with onset in infancy or early childhood, in which genetic and non-genetic influences acting either alone or in combination contribute to the development of ASD. According to the World Health Organization, ASD is characterized by impairments in social and communication skills, rigid or repetitive behaviors, atypical interests and differences in the perception of sensory stimuli [1]. The neurodiversity paradigm diverges from the traditional medical model’s understanding of autism as a disorder, viewing common traits in autism as neurological differences instead of deficits, thus shifting attention away from a disease model, while highlighting unique autistic strengths and embracing autism as a manifestation of neurological diversity that needs no cure [2]. 

Behavioral and psychiatric disorders in individuals with ASD are prevalent, and their impact is significant. A growing body of research reveals evidence of the frequent association between ASD and irritability, aggression, self-injurious behaviors, ADHD, anxiety, obsessive compulsive disorder, gender dysphoria, mood disorders, suicidality, substance use disorders, catatonia, psychosis and schizophrenia spectrum disorders. The symptoms associated with many behavioral and psychiatric disorders that commonly occur in individuals with ASD can overlap with core characteristics of ASD, which results in diagnostic challenges [3].

The risk for co-occurring behavioral and psychiatric disorders is influenced by individual differences including age, intellectual functioning, sex and genetic factors [4], with a majority of existing studies focusing on children and adolescents with ASD [5]. It has been shown that older autistic adults are at lower risk for having a co-occurring psychiatric diagnosis compared to their younger cohorts, a pattern similar to that found in the general population [6]. There is even less that is known about co-occurring conditions in autistic individuals with intellectual developmental disabilities. Although about one-third of people historically diagnosed with ASD have an intellectual disability [7], this subgroup is often excluded in the literature describing behavioral and psychiatric disorders in ASD [8].

ASD is a heterogenous genetic disorder with a 4:1 male-to-female ratio and over 800 ASD-related genes recognized with hundreds of chromosome aberrations, dozens of identified syndromes and a complex interplay between inheritance and environmental factors influenced by epigenetics [9,10,11]. Advances in understanding the role of genetics in human disease, particularly disorders of neurodevelopment, have been achieved by revolutionary changes in genomic technology with next-generation sequencing (NGS), computer program analysis and bioinformatics. This knowledge has created the opportunity for more detailed and rapid clinical evaluations and genetic testing options for patients presenting with neurodevelopmental disorders, intellectual disabilities and ASD. The ability to provide early diagnoses of inherited disorders has resulted in the development of clinical trials leading to new treatments for up to 3% of the world’s population with developmental disabilities, for whom there exist substantial comorbidities, high lifetime costs and the associated emotional burden of living with conditions that were previously considered to be untreatable [12,13,14,15,16]. The objective of this review is to demonstrate that advanced genomic and pharmacogenetic testing has the potential to improve both diagnostic evaluations and treatment interventions for individuals with autism. 

## 2. Behavioral and Psychiatric Observations in Autism

### 2.1. Behavioral and Psychiatric Conditions Often Associated with Autism 

#### 2.1.1. Irritability, Aggression and Self-Injurious Behaviors

Autistic individuals often experience elevated levels of irritability (e.g., temper tantrums, frustration or angry outbursts) and problem behaviors (e.g., physical aggression toward others, self-injurious behaviors or property destruction). Deficits in emotional self-regulation (e.g., using maladaptive emotion regulation strategies such as perseveration or shutting down) are common in autism and may result in anger being experienced more intensively, and in turn, strong negative emotions can trigger aggressive behaviors. Additionally, impairments in social cognition including inaccurate assessments or misinterpretation of social intent can promote aggressive behaviors [17]. 

Self-injurious behaviors (SIBs) are acts of physical harm inflicted upon oneself that have the potential to result in injury. Examples include hitting, pinching, scratching, biting, head banging and hair pulling. In contrast, stereotyped self-stimulatory behaviors typically involve repetitive or ritualistic movements, gestures, or vocalizations (e.g., repeated sounds, words or phrases) [18]. These behaviors can be persistent or episodic, spontaneous or repetitive, often without any identified cause, or may tend to occur within specific contexts or in response to certain triggers or situations. Risk factors for more serious and persistent SIBs include intellectual disability, limited communication skills, lower adaptive functioning, impairments in impulse control [19], sensory processing deficits or chronic sleep problems [20]. 

#### 2.1.2. ADHD and Executive Functioning Deficits

Attention Deficit Hyperactivity Disorder (ADHD) is characterized as a neurodevelopmental disorder manifested by core symptoms of distractibility and impulsivity, either with or without hyperactivity. In comparison to typically developing peers, limitations of executive functioning are common in both ASD and ADHD, suggesting that both may be conceptualized through an “executive functioning deficit model” [21]. Executive functions are cognitive control mechanisms that modulate perceptual experiences, motor responses, emotion regulation and behavioral reactions, as well as enable cognitive skills providing the ability to sequence, prioritize, plan, make decisions, anticipate novel situations, evaluate risks, utilize effective problem solving and develop adaptive coping strategies [22]. 

Developmentally appropriate social skills are commonly underdeveloped or impaired in both ASD and ADHD. The evaluation of social functioning involves an assessment of the individual’s social skills within the context of a particular interpersonal encounter. Social skills deficits in ADHD typically include a lack of attention to or disregard of situational or non-verbal cues, as well as intrusive or impulsive tendencies in communication and social interactions. In comparison, ASD related social challenges tend to manifest as either poorly developed or awkward social skills, with an apparent indifference to or lack of awareness for social cues, and for some individuals, this includes active avoidance of social engagement [23]. 

#### 2.1.3. Anxiety and Anxiety Disorders

Anxiety involves the expectation of a future threat, whether it is real or imagined. Anxiety disorders differ from what is considered to be age-related normal anxiety or fear as they cause significant distress, are functionally impairing, are excessive for the situation, or persist beyond developmentally appropriate periods. Leo Kanner [24], in 1943, described excessive anxiety in autistic individuals around unanticipated changes in the environment, alterations in schedules or unexpected events, viewing the behaviors of an autistic child as driven by an intense desire for maintaining sameness. His view was that difficulties with adapting occur in response to obsessive, restricted and perseverative features associated with autism. Rigid behaviors, including verbal rituals, compulsive routines (e.g., ordering and lining up objects), restrictive or rule-based preferences (e.g., eating only foods of one color), if interfered with or prevented, tend to trigger anxious distress. 

Among the anxiety disorders associated with ASD, the most common ones are specific phobias and generalized anxiety disorder, followed by social anxiety disorder and separation anxiety disorder [25]. It can be clinically challenging to diagnose specific anxiety disorders in the context of ASD due to shared features associated with both autism and anxiety disorders. For example, social anxiety disorder which is characterized principally by a fear of negative evaluation by others, and therefore, often leads to a lack of active engagement in social situations, which can manifest similarly to the core social communication deficit seen in autism [26]. Children with ASD often experience emotional and behavioral difficulties when they are overwhelmed with sensory stimulation. Deep pressure is a therapeutic modality utilized in occupational therapy to have a calming effect on the child by decreasing sympathetic arousal. An inflatable wrap called an autism hug machine portable seat has been shown to reduce neurobiological stress as measured by the sympathetic response, which in turn reduces problematic behaviors in children with ASD [27,28]. 

#### 2.1.4. Repetitive Behaviors versus Obsessive Compulsive Disorder

According to the *Diagnostic and Statistical Manual of Mental Disorders, fifth ed. (DSM-5),* the symptoms associated with ASD include restricted, repetitive patterns of behavior, interests or activities [29]. Repetitive behaviors in ASD can include recurrent vocalizations such as repeating certain noises, words or phrases, fixating on topics of special interest, rigid behaviors such as inflexible insistence on specific routines in everyday life, listening to the same song or watching the same videos repeatedly, or ritualistic stereotyped movements such as symmetric flapping or twirling of the hands, rocking of the body or the spinning of objects. Repetitive behaviors characteristic of ASD may appear to be very similar to the compulsive rituals often associated with obsessive compulsive disorder (OCD) [30]. 

OCD is characterized by a pattern of obsessive thoughts and compulsive behaviors that interfere with daily activities and cause significant distress. Obsessions in OCD consists of intrusive thoughts, either driven by an intense need for organization or symmetry, in response to fear of disease or contamination, or as an attempt to ward off unwanted urges or morally unacceptable thoughts or images that trigger significant anxiety. Compulsive behaviors in OCD are typically performed in response to intrusive thoughts in an unconscious effort to relieve distress. In comparison to OCD, repetitive behaviors in ASD are generally preferred, performed for the purpose of self-soothing, and do not typically cause distress. Repetitive behaviors in both OCD and ASD are often disruptive or time consuming and can lead to behavior problems including tantrums or aggression, particularly when others attempt to alter or interrupt the behavior [31]. 

#### 2.1.5. Gender Dysphoria

Gender dysphoria is caused by a misalignment between a person’s biological or birth-assigned sex and their personal experience of gender identity, whereas gender variance (or gender diversity) describes when an individual’s gender role and behaviors deviate from the culturally defined or socially expected gender norms [32]. Autistic individuals report a higher number of gender dysphoric traits compared to that of non-autistic peers [33]. Multiple studies report that ASD is over-represented in those with gender dysphoria and in gender-affirming specialty care clinics [34,35]. Strang et al. published clinical guidelines to help guide evaluation and treatment considerations for adolescents with co-occurring gender dysphoria and ASD [36].

Gender-diverse autistic adolescents and adults are known to have increased rates of mental health problems, including elevated risks for depression and suicidality compared to those of either autistic or transgender persons individually [37]. Due to associated social and communication differences associated with autism, they can face significant challenges in advocating for their gender-related healthcare needs. To optimize the success in patient outcomes, it is essential that medical providers become familiar with established standards of care for gender-diverse autistic individuals, help them to enlist family and social support, provide guidance in accessing gender-affirming healthcare, actively partner with other members of the patient’s care team to coordinate treatment interventions and obtain effective mental health consultation when it is needed [38]. 

#### 2.1.6. Depression and Bipolar Disorders

Major depressive disorder (MDD) consists of a predominantly depressed or irritable mood, accompanied by a loss of interest or pleasure in previously enjoyed activities, as well as a host of other distressing psychological and somatic symptoms, lasting for a period of at least 2 weeks [29]. An approximately two-fold risk for developing major depression in autistic young adults was reported in a population-based cohort study, which also found that MDD is more common in those with ASD than it is in their non-autistic siblings. ASD, particularly ASD without intellectual disability, is associated with elevated rates of major depression in young adulthood, and it is likely that the explanation is related to both shared genetic and environmental factors [39]. Identifying a major depression when it occurs in the context of ASD can be challenging given the overlap of common depression symptoms and features of ASD, as well as by a lack of evidence supporting the validity of commonly used assessment instruments for diagnosing MDD in autistic individuals [40]. 

Bipolar Affective Disorder (BAD) has two main subtypes. Bipolar I Disorder, which closely aligns with the classically described manic-depressive illness, involves both full-blown manic and major depressive episodes, and if the patient is not stabilized with effective treatment, it tends to evolve into a pattern of recurrent episodes that have the potential to escalate, with an associated risk for brief periods of psychosis. Bipolar II Disorder tends to follow a less severe course, defined by one or more episodes of major depression and hypomania [29]. BAD is more likely to occur in association with autism than it is in the general population. Not only is the risk of bipolar disorder greater for those with ASD compared to age- and sex-matched controls from the general population, but the risk is also higher in family members with ASD than it is in non-autistic siblings [41]. Diagnostic challenges exist because when BAD occurs in the context of autism, it may have an atypical presentation, thus causing either a lack of recognition or worse, an incorrect diagnosis, given that when psychotic symptoms are present, the clinical picture can be easily mistaken for schizophrenia [42].

#### 2.1.7. Suicidality

The elevated risk for suicidality in ASD has been historically under-recognized. The majority of studies evaluating suicidality in ASD have been conducted only in recent years [43]. A landmark study that assessed suicidality in a sizable clinic population of autistic adults formerly diagnosed with Asperger syndrome (an outdated diagnosis defining a sub-group of individuals with ASD with normal intelligence and greater impairment in social skills compared to those with other autistic traits) found that about two-thirds of them endorsed a history of suicidal ideation, with one-third having previously planned or attempted suicide [44]. Similarly, a population study in Sweden found that individuals diagnosed with ASD were significantly more likely to have suicide listed as the cause of death compared to the average proportion of suicide deaths reported in the general population [45].

The stress associated with “camouflaging” (a term used to describe those with ASD who intentionally mask their autistic traits or adapt their behavior with the goal conforming to neurotypical expectations) in autistic adolescents and adults contributes to increased stress, depression and suicidal behaviors in ASD [46]. Additional risk factors for suicidality in ASD include a history of behavioral problems, bullying, victimization, male gender, minority identity and lower socioeconomic status or educational attainment. Finally, the associated risks for suicidal thoughts and behaviors shared with the general population include recurrent self-injurious behaviors, psychiatric disorders and unstable employment, all of which occur at a greater frequency for individuals with ASD [47]. 

#### 2.1.8. Substance Use Disorders

An emerging body of research suggests that ASD is associated with double the risk for developing a substance use disorder (SUD). There is no clinical evidence explaining the lack of attention to the risk for SUD in ASD. The fact that SUDs were historically thought to be rare among individuals with ASD may be understood in considering the divergent culturally constructed narratives for the two conditions. The albeit naïve or simplistic ASD narrative portrays the individual as blameless and innocent, whereas the judgmental and moralistic narrative of SUD depicts the person as corrupt and unworthy, with the opposing narratives leading to cognitive discordance, and hence, bias. Clinicians need to remain vigilant when adopting the routine practice of screening individuals with ASD for SUD in the same manner they would for other patients and provide appropriate treatment for this common and potentially life-threatening co-occurring morbidity [48]. 

With regard to the susceptibility for developing SUDs, autistic individuals share common risk factors that exist in the general population including a genetic predisposition, environmental effects, stressful family events, early nicotine or other substance use, psychological distress and co-occurring emotional (i.e., depression) and behavioral conditions (i.e., ADHD, conduct disorder and anti-social personality disorder). Additional risk determinants that are more likely to be relevant in autistic individuals include low social support, dysfunctional coping strategies and poorer executive functioning. Social isolation compounded by social communication deficits may lead to difficulties with self-management, including a higher risk for establishing a pattern of substance abuse. The evidence suggests that autistic persons are at increased risk for developing substance use disorders, potentially via self-medicating with the goal of seeking relief from the multitude stressors they are commonly faced with or using drugs or alcohol to calm social anxiety or reduce the stress experienced in social interactions [49].

#### 2.1.9. Catatonia

Catatonia is a neuropsychiatric syndrome which has two subtypes. The akinetic (or retarded) type is the most frequent one and is characterized by a slowing or reduction of motor movements, immobility, rigidity, staring, mutism, withdrawal or refusal to eat. It may also include bizarre features such as posturing, grimacing, negativism, waxy flexibility, echolalia, echopraxia, stereotypy, verbigeration and automatic obedience. Hyperkinetic (or excited) catatonia is a less common presentation, in which there are prolonged periods of psychomotor agitation characterized by rigidity, autonomic dysregulation and altered mental status, and it can lead to life-threatening complications, including death, if it is not rapidly identified and treated [50]. 

There is an increased risk for catatonia in ASD, as well as other neurodevelopmental disorders, compared to that of the general population, occurring most often in adolescence and young adulthood [51]. Diagnostic challenges exist when catatonia develops in the context of autism given the overlap of behavioral features between the two conditions. Catatonic symptoms, including mutism, stereotypic speech, repetitive behaviors, echolalia, posturing, mannerisms, purposeless agitation and rigidity, can be mis-identified as core features of ASD [52]. Catatonia should be considered in autistic individuals when there is an obvious and marked deterioration in movement, vocalizations, pattern of activities, self-care and daily life skills. When accurately diagnosed catatonia can be medically treated in a timely manner, improved outcomes are demonstrated, and a full recovery is seen in most cases [53].

#### 2.1.10. Psychosis and Schizophrenia Spectrum Disorders

Psychosis is defined by a loss of reality orientation, with symptoms including hallucinations, paranoid or delusional thoughts (defined as fixed, false beliefs), and can occur as a result of a coexisting medical or psychiatric disorder. It is assumed that there is an underlying vulnerability to developing psychosis in ASD, which is likely related to overlapping genetic findings in ASD and primary psychotic (schizophrenia spectrum) disorders [54]. The risk factors for developing psychosis in ASD, when present, include major depressive disorder, anxiety disorders and the emotional trauma caused by bullying and other victimization, social bias, discrimination, unemployment, disability or other stressors commonly experienced by autistic individuals.

Schizophrenia is a serious and chronic mental illness that, in addition to psychotic symptoms, is defined by cognitive impairments, as well as persistently disordered ideas, beliefs, perceptions and behaviors [55]. Zheng et al. conducted a systematic review and meta-analysis of studies published over 10 years, demonstrating a significantly increased prevalence of schizophrenia in ASD [56]. The overlap in clinical features, which can superficially appear to be similar between ASD and schizophrenia, increases the possibility of a mistaken diagnoses. The idiosyncratic ideas of a person with ASD, for example, can be mistaken for the delusional beliefs which occur in psychosis. Likewise, cognitive rigidity and behavioral inflexibility in ASD can be interpreted as delusional in nature, potentially leading to a misdiagnosis of mental illness [57].

## 3. Genetics, Evaluation, Conditions and Genetic Testing in Autism

Advances in genetic technology and testing used to identify causation in patients with ASD have led to the identification of a specific etiology in 40% of patients presenting for genetic services using a three-tiered clinical genetics approach [58,59], in which genetic syndromes, molecular and cytogenetic defects and metabolic disturbances were evaluated. For example, mitochondrial disorders may account for up to 20% of individuals with ASD [60], along with involvement of other metabolic disturbances such as untreated PKU and Smith–Lemli–Opitz syndrome [11].

Children with ASD are also reported with microdeletions or duplications at the chromosome level for chromosome regions 1q24.2, 2q37.3, 3p26.2, 4q34.2, 6q24.3, 7q35, 13q13.2-q22, 15q11-q13, 15q22, 16p11.2, 17p11.2, 22q11, 2q13 and Xp22 [61]. Additional cytogenetic disorders have been found with new ultra-high-resolution microarray technology, including the emerging 15q11.2 BP1-BP2 deletions and duplications [10]. GWAS findings in ASD and broad autism phenotypes in extended pedigrees were studied in large cohorts in Canada and the United States showing other chromosome regions such as 1p36.22, 2p13.1, 6q27, 8q24.22, 9p21.3, 9q31.2, 12p13.31, 16p13.2 and 18q21.1 [62]. Further studies in ASD include the 15q11-q13 deletion (either of maternal origin as seen in Angelman syndrome or paternal origin in Prader-Willi syndrome), the 15q11.2 BP1-BP2 deletion (Burnside–Butler) syndrome or other chromosome 15 defects such as15q duplications or marker chromosome 15s. Recognized genetic disorders with chromosomal defects include Turner (45,X), Down (trisomy 21), Williams (chromosome 7q11.2 deletion), Smith-Magenis (17p11.2 deletion), Shprintzen/velocardiofacial (22q11 deletion) and Phelan-McDermid (22q13 deletion) syndromes. 

Single gene disorders which include both autosomal and X-linked defects associated with autism as a feature include tuberous sclerosis (*TSC1* and *TSC2* genes), neurofibromatosis (*NF1* and *NF2* genes), X-linked Rett (*MECP2* gene) and fragile X (*FMR1* gene) syndromes. Other syndromic conditions associated with autism include Sotos, Noonan, Moebius, Cohen, De Lange, Joubert, myotonic dystrophy and oculo-auriculo-vertebral spectrum, along with *PTEN* gene disturbances with extreme macrocephaly [11,63,64]. Finally, environmental factors known to contribute to ASD include parental age, perinatal factors, sex steroids, maternal health and maternal nutrition, as well as fetal exposure to drugs, toxins, alcohol, smoking, maternal diseases and infections [65]. Therefore, an environmental history should be obtained when one is pursuing potential contributing factors in the cause of autism when the patient is presenting for evaluation and testing. 

### 3.1. Genetic Considerations in Autism

#### 3.1.1. Genetics, Clinical Evaluation and Genetic Testing of ASD

ASD is on the rise around the world and reported at a higher rate than congenital defects such as brain malformations or for Down Syndrome are [66,67]. In 2018, the Center for Disease Control and Prevention reported that 1 out of 44 children (2.2% of 8-year-old children across 11 sites in the USA, based on health and education records) met the diagnostic criteria for ASD (https://www.cdc.gov/media/releases/2021/p1202-autism.html (accessed on 30 October 2022)). As a spectrum disorder, a wide range of associated clinical conditions and deficits have been noted, particularly involving emotional self-regulation and social skills, with as many as 30% of individuals with ASD not speaking. Verbal and non-verbal communication impairments are common in ASD and impact the development and maintenance of relationships. Differences involving the experience of intensity, regarding idiosyncratic or highly focused interests and unusual responses to sensory inputs, are generally noted beginning in early childhood, however a diagnostic evaluation for ASD may not be sought for an extended period of time. 

Autism is recognized as the most heritable neurodevelopmental disorder, with monozygotic twins having concordance rates that are about three times greater than those for dizygotic twins (0.98 and 0.53, respectively) [68,69]. Studies of multiple families and types of twinning indicate the importance of genetics in the causation of ASD. The best estimate for heritability is 80% reported in a review of more than two million individuals from six hundred and eighty thousand families from multiple countries indicating that the study of genomics can become a medical marker for ASD [70]. About 10% of individuals with autism are from families without a positive family history, referred to as simplex or sporadic autism, and are due to ASD gene defects including deletions or duplications found when they are studied at the chromosome level, particularly using chromosome microarray analysis. Microarrays are comprised of DNA probes for identifying copy number variants (CNVs) [71]. Individuals with autism and a positive family history are referred to as multiplex people. They are more likely to show more individual defects at the gene level (mutations) compared with those of typically developing children studied as controls, but not at the larger chromosome level with microarray analysis. The majority of CNVs are due to chromosomal deletions involving single or multiple gene conditions found in about 20% of individuals with ASD. Both genome-wide association studies (GWAS) and genetic linkage analysis have identified hundreds of DNA polymorphisms clustered in ASD risk gene loci recognized on all chromosomes and within the human genome, consisting of around 20,000 genes [9].

Chromosomal microarray analysis has demonstrated the highest diagnostic yield in individuals with ASD compared to those of other genetic tests, but advanced next-generation sequencing is continuing to identify other subtle genetic changes that have not been detected with microarray analysis. High-resolution microarrays utilize both CNV and polymorphic DNA probes to test for structural chromosome patterns such as deletions/duplications or heterozygosity patterns in patients with neurodevelopmental disorders including ASD. Microarrays are considered to be a first-tier genetic test and are best utilized to identify deletions or duplications of multiple or single genes that might be implicated at the chromosome level. For example, Ho et al. [10] in 2016 summarized the results of ultra-high-resolution microarray testing of patients presenting with neurodevelopmental disorders and/or autism for genetic services in a commercial laboratory setting. They used 2.8 million CNV and single nucleotide polymorphic (SNP) DNA probes optimized for neurodevelopmental disorders, reporting an overall CNV detection rate of 28.1% in 10,351 consecutive patients (average age 7 years, M:F ratio 2.5:1), with the overall detection rate for individuals with ASD being significant at 24.4%. In those with ASD, the most common cytogenetic abnormality was a 15q11.2 BP1–BP2 deletion (see Figure 1). Those with neurodevelopmental disabilities without ASD showed that the 22q11.2 deletion is the most common finding. Overall, 85 genetic findings were identified, with 9% of them having the 15q11.2 BP1-BP2 deletion, followed by 16p11.2 deletion at 5% and 16p11.2 duplication at 5%. Other findings included the 15q13.3 deletion, 16p13.1 duplication and *NRXN1* gene deletion, all at 4% (see Figure 2).

#### 3.1.2. Known and Putative Genetic Determinants of ASD

Our understanding of both the concept and causation of autism has exponentially improved since Leo Kanner’s first clinical descriptions in 1943 [24]. Butler et al. [9] searched the medical literature and identified about 800 clinically known, relevant or susceptible genes reported in autism, compiled a master list of genes for ASD identified in the literature with supporting evidence from peer reviewed medical resources by searching keywords related to autism and genetics and plotted their location on chromosome ideograms. The list of genes was arranged in alphabetical order in tabular form, with gene symbols plotted on high-resolution human chromosome ideograms to allow clinical and laboratory geneticists to access visual images of the location and distribution of ASD genes. For example, 74 genes were plotted on the X chromosome, which were compared to 54 genes on the similar size chromosome 6, thus chromosome X has about 50% higher gene frequency, supporting an X-linked affected male-to-female ratio of 4:1 for autism. These findings may inform diagnosis and gene-based personalized care to allow more accurate genetic counseling for at-risk family members. 

Mammalian or human chromosomes, over the course of evolution, have become substantially complex with the assembly and configuration status for a specific chromosome influencing gene expression by their chromatin structure and architecture. The configuration of each chromosome is based on its size, banding pattern and centromere position, which in turn may influence access, expression and function at the individual gene level. Genes are compartmentalized or packaged within chromosomes, and the architecture are identifiable by chromosomal staining, with light and dark banding patterns viewed under a microscope. Different standing regions or chromosome bands are designated as heterochromatic or darkly G-positive stained or euchromatic or lightly G-negative stained bands. The location of ASD genes in G-negative or G-positive bands was investigated by McGuire et al. [72]. Genes located in the heterochromatin regions may play a larger role in different biological processes than the genes located in euchromatic or active-gene rich regions on the chromosome do. Physical measurements of established chromosome banded ideograms were made based on chromosome size and appearance. Euchromatic G-negative band regions contain 60% of the known protein coding genes, while the remaining 40% are distributed across the G-positive heterochromatin bands. ASD genes were disproportionately over-represented in the darker heterochromatin sub-bands. This distribution and location of ASD genes may allow us to obtain a better understanding of neurodevelopment and function specifically associated with ASD genes and their locations on chromosomes impacting availability for gene expression [72]. 

Over the past 40 years, advances in genomics technology with bioinformatics and a rapidly growing interest in genetics research has led to discoveries of specific chromosomal and gene defects and dozens of recognized genetic syndromes associated with autism. For example, inborn errors of metabolism identified in individuals with autism include adenylate succinase deficiency, lactic acidosis and mitochondrial DNA defects or dysfunction. The mitochondria are organelles found in the cytoplasm that play an important role in adenosine 5′ triphosphate (ATP) production required for energy utilization through oxidative phosphorylation carried out by the electron transport chain made up of Complexes I, II, III and IV found in the inner membrane of the mitochondria, containing about 100 proteins [60,73,74,75]. These proteins are encoded by both mitochondrial DNA (mtDNA) and hundreds of interactive nuclear genes required for cellular energy influencing neurodevelopment and brain function. Several types of mitochondrial defects exist, including a depletion form with a reduced number of mitochondria per cell affecting biochemical reactions within the mitochondria and individual cells, particularly those requiring more energy to carry out their functions such as the brain and muscles [60,67,76,77]. In humans, from 100 to 10,000 separate copies of mtDNA are usually present per cell, while some cells such as germ-line ones may have many more copies required for energy production. Inborn errors of metabolism are recognized as the causes of enzyme deficiencies, leading to an accumulation of substrates or substances that can be toxic to the developing brain. 

#### 3.1.3. Molecular Genetic Characterization and ASD

During the last decade an enormous number of published molecular genetic reports on autism have utilized in genomic technology, bioinformatics and computational biology, including genome-wide association studies (GWAS) across the fields of medicine, psychiatry and the social sciences. This research led to the interrogation of genomic variants to search for links among specific gene findings, disorders, traits or phenotypes of affected individuals. Furthermore, bioinformatics and computational biology software interactive programs combined with human genomic datasets have led to the discovery of gene-gene–protein interactions, biological pathways and molecular functions and a further understanding of the role of genetics and biology in neurodevelopment. 

Early GWAS analyses have found a strong association with single nucleotide polymorphisms (SNPs) and cadherin genes, including neuronal cell adhesion molecules with over 100 genetic loci reported with ASD [78,79]. More recent data indicate that over 2000 human genes are implicated in intellectual developmental disability and ASD [80]. Many ASD genes are located on the X chromosome (>150 out of 900 X-linked protein coding genes) and involved in chromatin remodeling, synaptic function, neuronal signaling and brain neurodevelopment [81]. Of the 800 genes implicated as clinically relevant ones that are known or susceptible in ASD [9], many include members of the neuro-ligand, neurexin, cadherin, *GABA* receptors and *SHANK* gene families. Other genes encode neurotransmitters and their receptors, transporters, brain-derived hormones, oncogenes, signaling and ubiquitin pathway proteins, neuronal cell adhesion molecules and epigenetics [9,82,83]. 

One of the ubiquitin related genes is *UBE3A* (ubiquitin protein ligase E3A), a maternally expressed gene on chromosome 15 that causes Angelman syndrome, an imprinted disorder, and when it is mutated, it shows features of autism [84]. Therefore, gene expression patterns and correlation with features of autism were studied in individuals with chromosome 15 disorders (Angelman, Prader-Willi and chromosome 15 duplication syndromes), including *UBE3A* gene and *SNORD116*, a paternally expressed transcript or snoRNA that serves as a precursor to the antisense *UBE3A* gene for regulation of its activity. The reported findings suggested the presence of novel interactions between expression of *UBE3A* and *SNORD116,* with brain specific processes underlying motor and language impairments in autism and these chromosome 15 imprinted genetic disorders.

Next-generation sequencing now enables the accurate detection of mutations or gene variants at the whole exome or genome level by sequencing nuclear and mitochondrial DNA, and is, therefore, potentially more informative than structural chromosome microarrays are for single gene changes. Newer chromosomal SNP microarrays can identify chromosome abnormalities at levels approximately 100 times smaller than those seen with standard high-resolution chromosome methods developed in the 1980s. In terms of evidence, Shen et al. [85] reported genetic findings from more than 900 patients presenting with ASD using standard karyotype chromosome analysis, fragile X DNA testing (Fragile X Syndrome is recognized as the most common cause of familial intellectual disability and/or autism, primarily affecting males) and chromosomal microarrays. They found abnormal karyotypes in 2.2% of the patients, abnormal fragile X testing in 0.5% and deletions or duplications in 18.2%, including recurrent chromosome deletions or duplications for 16p11.2, 15q13.2-q13.3, 7q11 and 22q11.2, and more recently, 15q11.2 BP1-BP2 [10,11,86]. 

Whole exome sequencing (WES) has yielded results ranging between 9 and 30% [11,87,88] in individuals with ASD. For example, Aspromonte et al. [89] studied intellectual disability and autism with next-generation sequencing using a panel of 74 genes involved in molecular pathways and pathogenesis of both intellectual disability and ASD in 150 patients, achieving a 27% total diagnostic yield. One of the largest exome sequencing studies in autism to date was conducted by Satterstrom et al. in 2020 [90] in nearly 12,000 individuals with autism, including proband–parent trios and controls implicated 102 autism risk genes, finding a significant 3.5 fold increase in the de novo protein truncating variants in their study population. However, most of the gene variants found in other studies were of unknown clinical significance and were not reported in the medical literature as disease-causing ones or included in human genomic databases. These variants were rarely reported as pathogenic or likely pathogenic due to limitations in bioinformatics, computational prediction or the paucity of data in the literature. A recognized complex interplay exists between inheritance and environmental factors that contribute to autism and is impacted by epigenetic regulation of gene expression. With improvements in genomic technology, bioinformatics, computational predictions and the enlarging of human genomic databases, more useful information will be learned about the genetic causation of and single gene findings for ASD.

#### 3.1.4. Genetic Technologies Advance Our Understanding of ASD

A better understanding of how variations in implicated genes influence the presence of co-occurring conditions and drug response is needed, particularly as this information can help to guide personalized treatment approaches for ASD. Veatch et al. [91], in 2020, established a protocol to pinpoint ASD genes that are more likely to have clinically relevant variants by developing a functional annotation pipeline. They began with a list of all the candidate genes implicated in recognized manifestations of ASD utilizing databases that represent multiple lines of evidence including genes expressed in the human brain involved in ASD, their relevant biological processes and phenotypes reported in mice, whose products possess certain pharmacogenetic variation, have been targeted by pharmaceutical agents or directly interact with specific genes having variants recommended to be tested for by the American College of Medical Genetics (ACMG). Of 956 ASD implicated genes studied in the full set, 18 were flagged based on evidence from all the categories, with notably none of the prioritized genes represented among the 59 genes compiled by ACMG, and 78% with a pathogenic or likely pathogenic classified disease-causing variant. This ongoing research should rapidly prioritize potentially actionable results from genetic studies and, in turn, inform clinical decision support for personalized care based on genetic testing. 

Butler et al. [92] utilized early advances in genetic technology and a bioinformatics approach in a group of 20 females with autism (average age 7.7 years, range from 5 to 16 years) from multiplex families and compared them to simplex or sporadic families, indicating single gene involvement with structural DNA changes. Five of the twenty females had functional variants of X-linked genes (*IL1RAPL1, PIR, GABRQ, GPRASP2* and *SYTL4*) along with cadherin, protocadherin and ankyrin repeat gene families (*CDH6, FAT2, PCDH8, CTNNA3* and *ANKRD11*) and other related genes for neurogenesis or migration (e.g., *SEMA3F* and *MIDN*). More recently, a report using next-generation sequencing data and a meta-analysis of the literature compared ASD, epilepsy and intellectual disability identified in 103 published studies (ASD, N = 14; epilepsy, N = 72; intellectual disability, N = 21) across 32,331 individuals with diagnostic yields of 17.1% for ASD, 24% for epilepsy and 28.2% for intellectual disability [93]. The meta-analysis yielded a similar result to that of the whole exome study in the female ASD cohort reported earlier by Butler et al. [92]. More research is needed to identify the most commonly cited genes and their gene-gene–protein interactions with pathway analysis to introduce new avenues for therapeutic agents and treatment. 

Complex inheritance patterns exist in neuropsychiatric illnesses, indicating multiple genetic and environmental factors influencing disease risk and course. Hence, the GeneAnalytics computer program was used to conduct a pathway analysis and genetic profiling with predictors to characterize common or susceptible genes for ASD (792 genes [9], bipolar disorder (290 genes [94]) and schizophrenia (560 genes [95]). This program utilized an analytical approach to compare and rank score the number and nature of overlapping genes among disorders, gene disease associations and pathways, gene functions and tissue specificity, subdivided into categories (e.g., diseases, tissues or functional pathway). Twenty-three genes were found to be common among all three disorders and mapped to nine biological superpathways. These included circadian entrainment with ten genes, amphetamine addiction with five genes and sudden infant death syndrome with six genes. The program identified brain tissues with involvement in gene activity and measures for the three disorders involving the medulla oblongata with eleven genes, thalamus with ten genes and hypothalamus with nine genes. Six common genes were also found including *BDNF*, *DRD2*, *CHRNA7*, *HTR2A*, *SLC6A3* and *TPH2*. Interestingly, these overlapping genes impact serotonin and dopamine homeostasis and signal transduction pathways affecting mood, behavior and physical activity. Converging effects were also recognized in pathways that govern circadian rhythms with 10 genes found in common (*SLC6A3*, *GSK3B*, *HTR2A*, *MAOA*, *NOS1AP*, *PDE4B*, *TPH2*, *CACNA1C*, *CHRNA7* and *DRD2*) and supported by a known core etiological relationship that exists between neuropsychiatric illnesses and sleep disruption with hypoxia and associated central brain stem dysfunction [82,96]. 

In summary, a total of 106 phenotypes were mapped to the 23 overlapping genes, with 36 phenotypes in common including 18 genes having highly matched ranking scores. The highest matched phenotypes were behavioral despair (learned helplessness) involving four genes in common, hypoactivity involving seven genes, abnormal serotonin levels involving four genes and abnormal response to novel objects involving four genes. Additionally, identified phenotypes impact GABAergic neuron morphology, synaptic transmission and a response to hypoxia and a risk of death. Of the three disorders with overlapping genes matched to superpathways, the circadian entrainment category was the number one matched finding, followed by amphetamine addiction. For the biological processes category, the startle response was ranked as number one, followed by positive regulation of axon extension, cellular calcium ion homeostasis, synaptic transmission, dopamine catabolic process and axon guidance with synapse assembly [80]. Three functional pathways described in ASD as potentially involved were chromatin remodeling (e.g., *CHD7*, *MECP2*, *DNMT3A* and *PHF2*), the Wnt pathway (e.g., *CHDF8, PAX5* and *ATRX*), other signaling superpathways (e.g., *GPCR*, *ERK* and *RET*) and mitochondrial dysfunction [82,96].

Bipolar disorder and schizophrenia share certain neuropsychiatric and behavioral disturbances with ASD, including atypical styles of social interaction and communication, cognitive and perceptual differences and sleep disturbances. Multiple overlapping genetic and environmental influences may implicate the risk of disease, as demonstrated by an increased likelihood of either schizophrenia or bipolar disorder in first-degree relatives [95], and a 3.6 fold elevated risk in those diagnosed as children with ASD for developing schizophrenia as adults. Conversely, the prevalence of ASD in individuals diagnosed with schizophrenia is significantly elevated compared with population norms [97,98]. 

#### 3.1.5. Treatable Medical, Neurological/Neurometabolic Conditions, Intellectual Disabilities and ASD

A defective *PTEN* gene was reported in about 20% of individuals with autism and macrocephaly, a common finding in ASD [63]. *PTEN* is an important tumor suppressor gene reported to play a role in tumor growth, hamartoma disorders, overgrowth and cancer [98,99,100,101]. Gabrielli et al. [101] utilized the GeneAnalytics pathways and gene profiling computer program to examine shared autism and cancer genes, which is related to earlier research demonstrating that individuals with ASD had abnormal neuronal growth patterns with an overabundance of dendritic spine profiles. They found that 17% of the 800 ASD genes overlap with recognized cancer genes involving 371 superpathways for major cell signaling and metabolic disorders (e.g., *CREB*, *AKT* and *GPCR*), along with 153 gene ontology (GO) biological processes, 41 GO molecular functions and 145 matched phenotypes. Several of these pathways and molecular mechanisms proposed a cross-talk between the canonical Wnt pathway and the Notch signaling cascade with other disturbed genes and interactions [90,102,103,104]. These pathways may play a central role in a wide range of biological processes and when they are abnormal, they may compromise biological output or function contributing to neuropsychiatric disorders, cell growth and even malignancy in autism. 

There is a growing list of over 1400 recognized inherited neurological and neurometabolic conditions (Inborn Errors of Metabolism Knowledgebase (IEMbase). (www.IEMbase.org (accessed on 10 December 2020)) causing intellectual disability that may also impact ASD, as recently reviewed by van Konijnenburg et al. [16]. They queried databases include PUBMED, OMIM and Orphanet, identifying 116 treatable inherited metabolic disorders, finding 139 genes accounting for 116 disorders, for which the most frequent therapeutic interventions were pharmacological or nutritional based, with vitamins and trace element supplementation. Other treatment plans included organ or stem cell transplantation, enzyme replacement and gene-based therapies, with reported treatment effects including the improvement of clinical deterioration (62%), neurological manifestations (47%) and development (37%). 

The first tier of laboratory testing recommends the non-targeted metabolic screening of fasting blood samples to rule out disturbances of unexplained developmental delay, intellectual disability and ASD. These tests include those on fasting blood lactate, folate, ammonia, copper, amino acids, total homocysteine, acylcarnitines, very long chain fatty acids, transferrin N-glycan profiling and urine samples for organic acids, oligosaccharides, creatine, glycosaminoglycans and guanidinoacetate levels, accounting for 59% or 69 of the treatable metabolic disorders. Along with this first tier testing, genetic laboratory evaluation is recommended to include exome sequencing and mitochondrial DNA testing to target a growing list of causative genes contributing to the 116 disorders involving 139 genes, as well as targeting molecular results in 37 genes without biomarkers, accounting for 23 of the 116 recognized intellectual disability disorders. For these 23 disorders, no specific biomarker is currently available, and molecular testing would be required for 20% of the treatable disorders [16]. 

A second tier would target metabolic testing to identify 24 disorders, or 21%, including access to cerebral spinal fluid, enzymatic assays, or urine purines and pyrimidines. Treatable disorders are categorized as vitamin and cofactor metabolism (25%), amino acid metabolism (24%), complex molecular degradation (9%), neurotransmitters (8%), nucleic acid disturbances (6%), glycosylation defects (5%), energy substrate metabolism (4%), trace metals and elements (4%), carnitine, fatty acid and ketone body metabolism defects (4%), lipid metabolism (3%), mitochondrial cofactor biosynthesis defects (2%), other disorders mitochondrial dysfunction (2%), and carbohydrate metabolism, peptide and amine metabolism, endocrine metabolic disorders, mitochondrial DNA-related disorders and nuclear-encoded disorders of oxidative phosphorylation defects (all at 1% each). More than one third of the known treatable intellectual disability disorders included in the current literature were identified during the past eight years, with the majority of the effective treatments based on nutritional interventions considered to be relatively less expensive and more widely available than other options such as enzyme replacement are [16].

Examples of inherited metabolic disorders impacting neurodevelopment and function are numerous, with all modes of inheritance (e.g., autosomal dominant, autosomal recessive and X-linked). Clinical treatments have helped to halt or slow down disease progression (62%), impaired neurological manifestations (47%), systemic manifestations (44%) and psychomotor or cognitive development (37%), prevented acute metabolic decompensation (30%), improved seizure/epilepsy control (22%) and improved psychiatric disturbances (21%) [16]. Inherited metabolic disorders include disorders of glycosylation (e.g., SLC35A2-CDG); amino acid metabolism (e.g., carbamoyl phosphate synthetase 1 deficiency); complex molecule degradation (e.g., α-mannosidase deficiency); energy substrate metabolism (e.g., creatine transporter deficiency); carbohydrate metabolism (e.g., GLUT1 deficiency); fatty acid, carnitine and ketone body metabolism (e.g., mitochondrial acetoaceyl-CoA thiolase deficiency); lipid metabolism (e.g., 7-dehydrocholesterol eductase deficiency); nucleic acid metabolism (e.g., phosphoribosylpyrophosohate synthetase deficiency); trace metals (e.g., Wilson disease); vitamin and cofactor metabolism (e.g., methylmalonic aciduria); mitochondrial myopathy; neurotransmitter disorders (e.g., tyrosine hydroxylase deficiency).

An example of an autosomal dominant genetic disorder associated with an increased risk of developing autism, for which there are potentially effective treatment options, is tuberous sclerosis, which is caused by defects of the *TSC1* or *TSC2* genes [101,105]. Tuberous sclerosis is a classical genetic disorder with neurodevelopmental problems, seizures, renal angiomyolipomas, astrocytomas, skin depigmentation, ADHD and autism (www.omim.org OMIM #191100). Kilincaslan et al. [105] reported on the beneficial effects of everolimus medication on autism and ADHD in patients with tuberous sclerosis. This drug inhibits mTOR, one of the top thirty biological super-pathways identified when comparing shared autism and cancer genes [101]. The effects of this drug on neuropsychiatric findings in those with autism require more testing. 

#### 3.1.6. Pharmacogenetics with Medication Management and Selection

Precision or personalized medicine is an emerging tool for use in clinical practice, which leads to improved medication selection and management based on an individual’s DNA pattern or pharmacogenetics [106]. The study of pharmacogenetics is based on structural DNA variation at the individual level that impacts the drug metabolism and therapeutic response based on the cytochrome P450 liver enzyme system, which is coded by individual genes throughout the human genome. Cytochrome P450 enzymes metabolize medications by the liver and influence drug concentrations in the blood. Most prescription drugs are metabolized by this enzyme system, and therefore, it plays a significant role in treatment success. The identification of specific cytochrome P450 gene polymorphisms can explain the source of variability in drug dosing and response, thus helping to guide effective treatments for individuals with ASD. 

There are over 50 cytochrome P450 liver enzymes that metabolize endogenous and xenobiotic substrates, including environmental pollutants and plant-based chemicals. These enzymes are also involved in the biosynthesis and metabolism of steroids, vitamins, hormones, lipids and prostaglandins [107,108,109,110,111,112]. About 90 percent of all drugs are metabolized by seven different liver enzymes including *CYP1A2*, *CYP3A4*, *CYP3A5*, *CYPC19*, *CYP2D6*, *CYP2C9* and *CYP2B6* [104]. The most common medications used to treat patients with psychiatric or behavioral problems including ASD are metabolized by *CYP2D6* [106,107,108,109]. Often drugs are metabolized by more than one cytochrome P450 enzyme, and some drugs such as risperidone require them to be broken down from an inactive drug or compound to generate an active metabolite or functional agent for treatment. Individuals who are either fast or slow metabolizers based on the cytochrome P450 system DNA patterns who take psychotropic drugs may respond differently to the recommended dosage, thereby producing adverse side effects or a lack of response. A better understanding of metabolic differences that occur with age and increased use of medications may further impact drug dosage and the selection of specific drugs for treatment. 

Clinical trials investigating new classes of drugs or advances being achieved in research on existing drugs for new purposes, including the treatment of behavior problems in ASD, are ongoing and hold promise for the discovery of improved therapies. Other important factors that may also play a role in the efficacy of medications in treating diseases or disorders include neurotransmitter receptors, cell membrane transporters and sensitivity markers, which vary from person to person and are coded by individual genes outside of the cytochrome P450 enzymes [107].

There is growing evidence that the cytochrome P450 enzymes may be altered by drugs or other substances that act as either inducers or inhibitors, as well as drug–drug interactions. Known inhibitors or inducers may include common sources in the diet such as grapefruit, broccoli, cabbage, cauliflower or coffee. All potential sources of inducers or inhibitors should be considered, along with knowledge of the individual’s enzyme activity. For example, reduced cytochrome P450 enzyme activity could be positively impacted by an inducer (a food source or drug) and increase the enzyme response in breaking down or metabolizing a drug prescribed to treat the patient, thus altering the therapeutic response. The person’s individual DNA pattern that encodes the liver enzymes involved in metabolism to generate a fast or slow metabolic rate regarding drug break down and treatment response can either negate the potential benefit of drug therapy and/or lead to adverse side effects. Drug–drug interactions and plasma drug concentration adjusted by half-life in accordance with response to inhibitors and/or inducers can also impact the medication levels and treatment success [105,106,107,108,109]. 

About 25 to 50% of individuals do not respond as they are expected to, to recommended drug dosages or other therapeutic treatment interventions, and this scenario also applies to those with ASD. The discovery of new classes of medications and research on existing drugs for new purposes to treat behavioral, psychiatric and medical conditions in patients with ASD are under investigation, with the goal expanding the treatment options and improving treatment outcomes. In addition, the application of genetic advances that improve our understanding of pharmacogenomics and gene-gene–protein interactions along with pathway and functional analysis may help to lead to new discoveries and applications for therapeutic interventions in ASD [111].

## 4. Limitations

The limitations noted include a relative lack of data in the literature regarding behavioral and psychiatric co-occurring conditions for adults, and particularly for older adults with ASD, as well as for individuals with the dual diagnosis of ASD and intellectual developmental disabilities. The authors additionally note that the currently existing genomic technology, bioinformatics and computational predictions, as well as the size of available human genomic databases and published information, limit what can be learned about genetic causation and single gene findings in ASD. Finally, it is recognized that there are limitations in the available research to date in identifying relevant genes, their variants and gene-gene–protein interactions, as well as the clinical utility of existing pharmacogenomics testing, with improvements being needed to guide the selection of effective therapeutic agents and other treatment interventions. 

## 5. Conclusions

In this review, we discussed the behavioral and psychiatric conditions commonly associated with autism, including irritability, aggression, self-injurious behaviors, ADHD, anxiety, obsessive compulsive disorder, gender dysphoria, mood disorders, suicidality, substance use disorders, catatonia, psychosis and schizophrenia. Genetic associations, identified syndromes and chromosomal defects including deletions and duplications, and gene variants associated with genetic contribution and high heritability estimates seen in ASD were described and illustrated. The largest high-resolution chromosomal microarray analysis of patients presenting with ASD for genetic laboratory services was 15q11.2 BP1-BP2 deletion (Burnside–Butler) syndrome as the most frequent finding, followed by 16p11.2 deletion, accounting for a combined 14% of the 85 genetic defects [10,11]. 

Advanced genomic testing including exome sequencing and computational analysis found several disturbed pathways and molecular mechanisms including chromatin remodeling, Wnt, Notch and other signaling superpathways, *PTEN*, *BDNF*, *NRNX1* and *SHANK3* and related genes involved in neurogenesis, metabolic problems and mitochondrial dysfunction. The genetic data analyzed using computer based programs found share mechanisms that may lead to the identification of common pathologies, disturbed gene and protein pathways, biological processes and molecular functions. This information may lead to a better understanding of causation with potential treatment options to lessen the severity of ASD-related challenging behaviors. 

Finally, the authors describe the role of pharmacogenetics and testing, identifying and defining variables impacting the metabolism of psychiatric medications for those with ASD, thereby helping to guide therapeutic pharmacological interventions. More studies are needed to further characterize the epiphenotypes within the autism spectrum, the contribution of gene variants, chromosome aberrations, and identified syndromes with potential treatable outcomes, with the goal of reducing comorbidities impacted by pharmacogenetic patterns, neurometabolic disturbances and environmental factors. More research could assist in earlier diagnoses, expand our understanding of causation and lead to improved, tailor-made therapeutic interventions for each affected individual.

## Figures and Tables

**Figure 1 genes-14-00677-f001:**
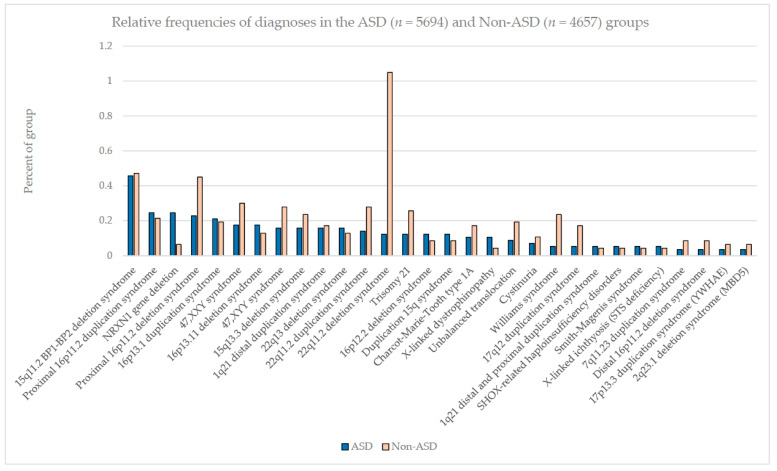
The relative frequencies of diagnoses in the combined ASD (n = 5694) and non-ASD (n = 4657) patient cohorts presenting for genetic services and laboratory testing using ultra-high-resolution chromosomal microarray analysis (reprinted with permission from Ho et al. [10]).

**Figure 2 genes-14-00677-f002:**
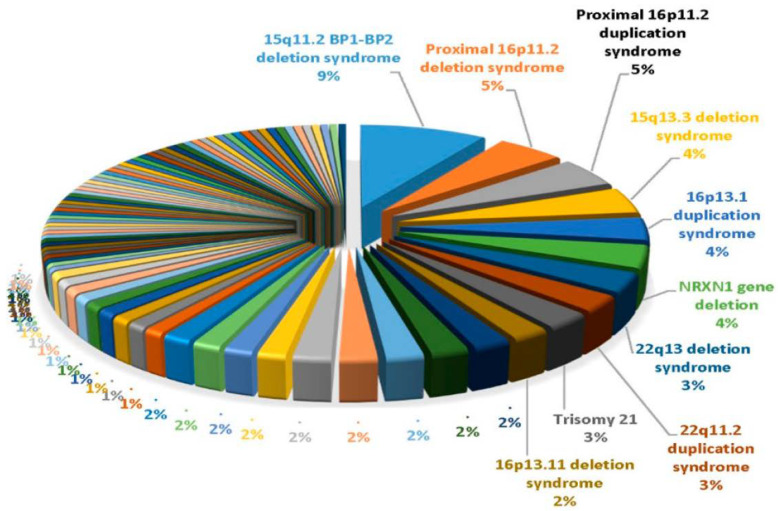
Pie chart showing the top ten genetic findings using ultra-high-resolution chromosomal microarrays from a large cohort of over 10,000 consecutive patients presenting for genetic services and laboratory testing with neurodevelopmental disorders affecting brain function and/or structure problems of unknown cause with developmental/intellectual disabilities and/or ASD with data previously summarized by Ho et al. [10] (reprinted with permission from Genovese and Butler [11]).

## Data Availability

No available data to share.

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
