# Peer review of "The Autism Spectrum: Behavioral, Psychiatric and Genetic Associations"

_genes, 2023, doi:10.3390/genes14030677_

Round 1

Reviewer 1 Report

1.      Please conclude your abstract with a "take-home" message.

2.      Put the keywords in a new order based on alphabetical order.

3.      It is encouraged not used abbreviations in the keywords section.

4.      Describe the novelty of the review article made by the author? From the results of my evaluation, it seems that many similar published works adequately explain what you have raised in the current manuscript. If there is something others really new in this manuscript, please highlight it more clearly in the introduction section.

5.      In order to demonstrate the review gaps that the current study aims to address, previous review linked to it need to be explained in the introduction part, including their work, their novelty, and their limitations.

6.      Please note that, last paragraph of the introduction section in this article should be explained the present objective.

7.      In line 99 section 2.1.3. Anxiety and Anxiety Disorders, the authors needs to extend explain anxiety research in children with autism. Some behavioural and studies related would help to improve that manuscript, and encouraged to incorporated as follows: DOI: 10.3390/bioengineering9040157 and 10.3390/bioengineering9020048.

8.      The authors need to improve the discussion in the present article become more comprehensive. The present form was insufficient.

9.      The present review's limitations should be added before moving on to the conclusion section.

10.   Mention further research in the conclusion section.

11.   Literature from the last five years should be enriched to reference. MDPI reference is strongly recommended.

12.   The manuscript needs to be proofread by the authors since it has grammatical and language issues.

13.   It is mandatory to provide a graphical abstract after revision is submission system.

Reviewer 2 Report

The authors presented a nice review manuscript entitled "The Autism Spectrum: Behavioral, Psychiatric and Genetic Associations". The content is of high quality and can be readily accessed and understood my general neurologists, pediatricians and geneticists. Few aspects need further assessment by the authors: 

1. Genes are presented in the text (both in the text and in graphs/pictures) without the use of italics. As the manuscript discusses aspects about human genetics, it is recommended to use italics in all descriptions of gene symbols in the texts (including in Abstracts). 

2. I suggest in the Introduction of the manuscript the possibility of adding content about the consequences (individual and population views) in the context of ASD (for example, describing aspects about costs, frequency, incidences in general or specific populations). 

3. Similarly to what was done in the subitem 3.1.5. (highlighting the importance of pharmacogenetics), it is interesting if the authors consider adding a special subitem (briefly) discussing the importance of genetic evaluation to make it possible the identification of potentially treatable neurological conditions, especially inherited neurometabolic disorders. One article example presenting similar content is discussed in: Orphanet J Rare Dis 16, 170 (2021). 

Round 2

Reviewer 1 Report

I am recomending the present manuscript to be accepted.